# Loss of CITED1, an MITF regulator, drives a phenotype switch *in vitro* and can predict clinical outcome in primary melanoma tumours

Jillian Howlin[1,2], Helena Cirenajwis[1], Barbara Lettiero[1], Johan Staaf[1], Martin Lauss[1], Lao Saal[1], Åke Borg[1], Sofia Gruvberger-Saal[1,3] and Göran Jönsson[1,3]

[1] Division of Oncology-Pathology, Lund University Cancer Center/Medicon Village, Scheelevägen, Lund, Sweden
[2] Cell and Experimental Pathology, Department of Laboratory Medicine Malmö, Lund University, Sweden
[3] These authors contributed equally to this work.

## ABSTRACT

CITED1 is a non-DNA binding transcriptional co-regulator whose expression can distinguish the 'proliferative' from 'invasive' signature in the *phenotype-switching* model of melanoma. We have found that, in addition to other 'proliferative' signature genes, CITED1 expression is repressed by TGF$\beta$ while the 'invasive' signature genes are upregulated. In agreement, CITED1 positively correlates with MITF expression and can discriminate the MITF-high/pigmentation tumour molecular subtype in a large cohort (120) of melanoma cell lines. Interestingly, CITED1 overexpression significantly suppressed MITF promoter activation, mRNA and protein expression levels while MITF was transiently upregulated following siRNA mediated CITED1 silencing. Conversely, MITF siRNA silencing resulted in CITED1 downregulation indicating a reciprocal relationship. Whole genome expression analysis identified a phenotype shift induced by CITED1 silencing and driven mainly by expression of MITF and a cohort of MITF target genes that were significantly altered. Concomitantly, we found changes in the cell-cycle profile that manifest as transient G1 accumulation, increased expression of CDKN1A and a reduction in cell viability. Additionally, we could predict survival outcome by classifying primary melanoma tumours using our *in vitro* derived 'CITED1-silenced' gene expression signature. We hypothesize that CITED1 acts a regulator of MITF, functioning to maintain MITF levels in a range compatible with tumourigenesis.

## INTRODUCTION

CITED1 is the founding member of the CITED (**C**BP/p300-**i**nteracting **t**ransactivator with glutamic acid [**E**]/aspartic acid [**D**]-rich C-terminal domain) family of transcriptional co-regulators and was originally cloned from a differential display screen between pigmented mouse B16 melanoma cells and their dedifferentiated weakly-pigmented derivative,

Corresponding author
Jillian Howlin,
jillian.howlin@med.lu.se

B16F10s. This led to speculation that CITED1 or *msg1* (melanocyte specific gene 1) as it was known at that time, was involved in the process of pigmentation (*Shioda, Fenner & Isselbacher, 1996*). Subsequently, *Nair et al. (2001)* reported that stable overexpression of CITED1 increased the levels of tyrosinase, dopachrome tautomerase (Dct) and melanin in B16 cells, reinforcing the idea that it had a role in melanogenesis. By 2005, as gene expression profiling became relatively commonplace, CITED1 was identified in several new screens of tumours and cell lines: two studies identified CITED1 as a gene whose expression distinguished nevi from primary melanoma, another found CITED1 to be upregulated in advanced stage melanomas in comparison to benign nevi or melanoma *in situ*, while expression profiling of an *in vitro* progression model identified CITED1 among a signature of genes lost in aggressive melanoma lines relative to primary melanocytes in culture (*Ryu et al., 2007*; *Haqq et al., 2005*; *Talantov, 2005*; *Smith, Hoek & Becker, 2005*).

Based on extensive gene expression profiling of melanoma cell lines *in vitro,* Hoek et al. proposed the 'phenotype-switching' model of melanoma that was independent of the degree of transformation or disease progression, and sought to explain the observation that melanoma cells altered between two states: those with high proliferative potential that are less invasive and those with high metastatic potential that are less proliferative. These separate but alternating states are controlled by different transcriptional programs and can be defined by specific gene signatures (*Hoek et al., 2008a*). MITF expression and many of its known targets (TYR, MLANA) define the 'proliferative' group, while the 'invasive' signature group is characterized by expression of negative regulators of the Wnt signalling pathway (WNT5A, DKK1, CTGF). CITED1 expression was associated with the proliferative pathway signature and subsequently confirmed in an updated and expanded data set to be significantly correlated with the proliferative phenotype ($P < 1.00E-05$, http://www.jurmo.ch/hopp, accessed 19 March 2013) (*Hoek et al., 2006*); (*Widmer et al., 2012*).

Studies on CITED1 suggest that it is a non-DNA binding nuclear transcriptional co-regulator capable of influencing TGF$\beta$ induced transcription mediated by ligand-induced SMAD hetero-oligomerization; estrogen-dependent transcription mediated by ER$\alpha$, and Wnt/$\beta$-Catenin-dependent transcription. These effects are dependent on CITED1-CBP/P300 binding via the conserved CITED family CR2 domain and while CITED1 is thought to act by stabilizing the CBP/P300-ER$\alpha$ interaction, in the case of $\beta$-Catenin it acts to repress transcription by competing for binding with CBP/P300 transcriptional co-activators (*Shioda et al., 1998*; *Yahata et al., 2001*; *Yahata et al., 2000*; *Plisov, 2005*).

Microphthalmia-associated transcription factor, MITF, acts as a master-regulator of melanocyte differentiation and as a result has been intensely studied in the field of melanoma research (*Widlund & Fisher, 2003*; *Levy, Khaled & Fisher, 2006*). It is a basic helix-loop-helix leucine zipper transcription factor that recognizes E-box and M-box sequences in the promoter regions of its target genes. Highlighting its importance in the disease, amplification of MITF locus has been found in >15% of metastatic melanomas and germline mutations in MITF that predispose carriers to melanoma development have also been found (*Garraway et al., 2005*; *Bertolotto et al., 2011*; *Yokoyama et al., 2011*). In melanoma cells the target genes of MITF include most notably TYR, MCIR, DCT,

MLANA involved in the process of pigmentation; cell cycle regulators such as CDK2 and CDKN1A and the more recently identified BRCA1 gene that has, with other target DNA repair genes, defined a role for MITF in the DNA damage response (DDR) (*Strub et al., 2011*; *Beuret et al., 2011*; *Giuliano et al., 2010*).

The regulation of MITF is complex and tightly controlled, exhibiting both transcriptional and post-translational regulation. There are several transcript isoforms, of which MITF-M is the dominant form expressed in melanocytes. Multiple signalling pathways converge on the MITF-M specific promoter that harbours binding sites for PAX3, SOX10, CREB, FOXD3, LEF-1 and BRN2 among other transcription factors (*Yokoyama & Fisher, 2011*; *Levy, Khaled & Fisher, 2006*). Additionally, the MITF target gene CDKN1A/P21 has been shown to act as reciprocal transcriptional cofactor independently of its CDK inhibitor function, suggesting the existence of at least one positive feedback loop (*Sestáková, Ondrusová & Vachtenheim, 2010*).

MITF post-translational activity can be affected by phosphorylation, sumoylation, ubiquitination and by binding with proteins that block access to the DNA binding domain such as PIAS3 (*Yokoyama & Fisher, 2011*; *Levy, 2001*). Oncogenic BRAF (but not wildtype BRAF), which is mutated in up to 50% of melanomas, also regulates MITF via simultaneously stimulating MITF activation through ERK phosphorylation, which leads to its subsequent degradation, and by inducing transcription of MITF via BRN2 upregulation (*Davies et al., 2002*; *Wellbrock et al., 2008*).

The consensus regarding why the cell invests such effort in maintaining control of MITF levels and why there are so many regulatory mechanisms, is that melanocytes and melanoma are exquisitely sensitive to even small variations in MITF expression. Ultimately its activity must be sustained within the narrow window permissive for continued survival and proliferation. In this study, we characterise the role of CITED1 as a novel regulator of MITF in melanoma.

## MATERIALS AND METHODS

### Cell lines

Cell lines were obtained from ATCC. HT144 and SKMEL3 cells were cultured in McCoy's5A supplemented with 10% and 15% foetal bovine serum (FBS), respectively. A2058, WM852 and WM239 were cultured in RPMI 1640 supplemented with 10% FBS; A375 and HMBC cells were cultured in DMEM supplemented with 10% FBS and SKMEL5 cells were cultured in MEM media supplemented with 10% FBS. Cells were grown in the presence of penicillin and streptomycin (50 I.U./mL) purchased from Invitrogen. Cell media and FBS were purchased from ThermosScientific, HyClone range. As of March 2014, these are part of the Life Technologies (Thermo Fisher Scientific) product portfolio.

### Gene expression analysis

RNA was isolated (4 replicates for each treatment) using a Qiagen RNeasy Plus mini-kit and the quality determined using a Bioanalyser (Agilent). Replicates were cell samples from separate wells, but plated on the same day and derived from the same passage

number. Gene expression experiments were performed using the Illumina HT-12 array covering more than 47,000 transcripts and known splice variants across the human transcriptome. The raw data was quantile normalized and Illumina control probes were removed from subsequent analysis using BASE (*Vallon-Christersson et al., 2009*). The data were exported to MeV, log2 transformed and gene and sample centred (*Saeed et al., 2003*). SAM (significance of microarray analysis) was performed using a two-group comparison; for the siRNA experiment the groups corresponded to siNEG vs. #1 & #3 siCITED1 and for the TGF$\beta$1 experiment the groups corresponded to cells with or without TGF$\beta$1 treatment. In both cases there was a median false discovery risk of 10 false-positive transcripts. Hierarchical clustering was performed to visualize the data. 1009 probes were significantly altered by the TGF$\beta$1 treatment while 312 probes were found to be significantly altered in the siRNA experiment (208 upregulated and 104 downregulated). These data can be found in Files S1 and S2, respectively. DAVID was used to assist in functional annotation of the final gene lists (*Huang et al., 2007*).

For the publically available data cited, 120 melanoma cell lines from three cohorts (*Johansson, Pavey & Hayward, 2007*; *Hoek et al., 2006*; *Greshock et al., 2010*) analysed by Affymetrix gene expression microarrays were collected, individually MAS5 normalized, and merged into a single cohort. Probe sets were collapsed into single genes and mean-centred across the entire cohort. These 120 cell lines and their associated normalized expression data can be found in Data S2. Data from Harbst et al. were classified using nearest centroid and Pearson correlation. Survival analysis and multivariate cox regression methods were performed in R.

## Transient transfections, promoter-reporter assay and TGF$\beta$1-treatment

Transient transfections were performed using Lipofectamine2000 and Opti-MEM reduced serum media (Life Technologies, Waltham, Massachusetts, USA) according to the manufactures recommendations. The siRNA was purchased from Applied Biosystems and the notations in the text: siNEG, #1 siCITED1 and #3 siCITED1 correspond to the catalogue ID numbers #4390843, #s8965 and #s224062 respectively. For the MITF targeting siRNA; N, siM1 and siM3 correspond to the catalogue ID numbers #4390843, #s8790 and #ss8792, respectively. For the luciferase reporter assay, a Dual-Luciferase Reporter assay system #E1910 (Promega, Southampton, UK) was used to measure relative reporter activity on a FLUOstar Omeaga microplate reader (BMG Labtech, Offenburg, Germany). A375 cells were transfected with a luciferase reporter construct harbouring 2.3kb of the MITF-M specific promoter in a PGL2 vector (*Wellbrock et al., 2008*). A pRL-Renilla Luciferase reporter vector was used as a control for each transfection. CITED1 was overexpressed using a pRc/CMV containing a N-terminal HA-tagged human CITED1 (transcript isoform 1) referred to as pCITED1 in the text (*Shioda, Fenner & Isselbacher, 1996*). An empty CMV-promoter expression plasmid, pcDNA3.1 (+) was used a negative control. Recombinant human transforming growth factor-$\beta$1 (TGF$\beta$1), #PHG9203 was purchased from Invitrogen. For the A2058 gene expression experiment, cells were exposed to either 5 or 10 ng/ml TGF$\beta$1 in serum-free media for 24 h. In the case of the Luciferase

reporter assay, cells were serum starved the day after transfection for 3 h and exposed to 5 ng/ml TGFβ1 in serum free media for 24 h prior to harvesting.

## Antibodies and immunoblotting

The following antibodies were used: anti-CITED1, #AB15096 from Abcam; anti-MITF (C5 clone), # MA5-14146 from ThermoScientific; anti-MITF (D5 clone) from Dako, #M3621, (used in Fig. 4C); anti-CDKN1A/P21, #2947 and anti-CDKN1C/P57, #2557 were purchased from CellSignaling Technology and anti-$\beta$-Actin (AC-15), #A5441 from Sigma-Aldrich. Cell lysates were resolved by SDS-PAGE (pre-cast gels purchased from Life Technologies) and transferred to 0.45 µm PVDF membranes purchased from Millipore by electroblotting. The membranes were blocked in 5% non-fat milk in TBST prior to incubation with primary antibodies diluted 2.5% non-fat milk. The blots were probed with the appropriate secondary antibodies (Pierce Biotechnology, Waltham, Massachusetts, USA) in 5% non-fat milk. The membranes were developed using ECL (GE Healthcare, Buckinghamshire, UK).

*A note on reproducibility*: Western blots were performed for a variety of purposes; in cases where it was simply a control to establish if siRNA downregulation had been successful, or confirm that an observation seen on mRNA level was also reflective of protein level, then there may have been only one Western blot performed on those particular samples (although often technical repeats were performed). In contrast, observations that were key to the hypothesis presented were performed a number of times to ensure reproducibility e.g., The effect of siCITED1 on upregulation of MITF and CITED1 overexpression on MITF downregulation was seen on samples for multiple experiments performed (i.e., >3): protein samples were used to confirm the initial observations from the GEX (using two siRNAs at two different timepoints) and two transfection of different amounts of pCITED1 plasmid; subsequently, two sets of timecourse experiments were performed to establish the effect independently, at both mRNA and protein level. The effect on P57 was initially observed using transfection of two different amounts pCITED1 plasmid and subsequently confirmed at multiple timepoints in both of two separate sets of timecourse experiments. The P21 effect was observed at single timepoint following siCITED1 treatment in both of two separate timecourse experiments. For the reciprocal effect of siMITF on CITED1 it should be noted that the effect was observed in 3 different cells lines, using at least two separate transfections.

## Cell cycle analysis

Flow cytometry was performed on a FACSCalibur (BD Biosciences, San Jose, California, USA) and subsequently analysed using ModFit (Verity House Software, Topsham, Maine, USA). Briefly, following transfection, confluent cells were detached, washed in 1XPBS and fixed in 70% ethanol. Prior to analysis they were stained with a propidium iodide solution and a 20G syringe was used to obtain a homogenous single cell solution. All events were saved (up to 20,000 events per replicate) ungated, using BD Cell Quest and the data exported to ModFit where following selection of the appropriate ploidy status, a standard auto-analysis fit using autolinerarity was performed. We found that a 2-cycle

aneuploid-dip/tetraploid was appropriate for HT144 and A2058 while 1-cycle diploid was suitable for A375.

## Alamar blue assay

The Alamar blue assay reagent was purchased from Invitrogen (Life Technologies, Waltham, Massachusetts, USA) and used according to the manufactures' instructions. Briefly, following transfection cells were seeded into 96-well plates at 5,000 cells/well. In each experiment, for each of the treatments i.e.: siNEG, #1 siCITED1 and #3 siCITED1, 8 wells spread over 3 rows were used. At the indicated time points (4, 72, 96 and 120 h post-transfection), Alamar blue was added and the cells incubated at 37 °C for 2 h. Fluorescence was measured (544 nm) on a FLUOstar Omeaga microplate reader (BMG Labtech, Offenburg, Germany). The values obtained at the 4-h time point were used to normalize the fluorescence readings to account for any initial cell counting error. Cells were also seeded in parallel for Western blot analysis (72, 96, 120 h) to ensure successful CITED1 down regulation.

## Droplet digital PCR

RNA was isolated from cells using a Qiagen RNeasy Plus mini-kit and quantified using a Nanodrop spectrophotometer (Thermo Fisher Scientific, Waltham, Massachusetts, USA). cDNA was generated from 50–100 ng total RNAs using 'iScript Advanced cDNA synthesis for RT-qPCR' (Bio-Rad). Bio-RAD's 'ddPCR Supermix for Probes' was then used with predesigned TaqMan gene expression assays (Applied Biosystems, Carlsbad, California, USA) consisting of specific primers and FAM labelled probes for MITF (#Hs01117294_m1), MITF-M isoform specific transcript (Hs00165165_m1)*, CITED1 (#Hs00918445_g1) and IPO8 (#Hs00183533_m1). (*There appeared to be no advantage in using the MITF-M isoform specific transcript over the MITF probe that could measure multiple isoforms). A manual cut-off for positive/negative droplets was selected using the Bio-Rad QuantaSoft$^{TM}$ data analysis suite to calculate the relative copies/µl of each transcript.

## RESULTS

### TGF$\beta$ induces expression of the invasive signature genes while suppressing a cohort of proliferative signature genes including CITED1

*Hoek et al. (2006)* noted that many of the genes that defined the invasive phenotype were commonly TGF$\beta$-driven while at the same time only the proliferative signature phenotype cells were sensitive to TGF$\beta$ growth inhibition *in vitro*. That MITF levels decrease and invasiveness is enhanced in response to TGF$\beta$ stimulation was also confirmed subsequently (*Pierrat et al., 2012*; *Pinner et al., 2009*). In agreement, we showed that the melanoma cell line A2058 upregulates WNT5A in response to TGF$\beta$ exposure and that exogenous Wnt-5a in turn, increased their invasive potential (*Jenei et al., 2009*). For the present study, in an effort to examine what other phenotype specifying genes were directly regulated by TGF$\beta$, we performed gene expression analysis and found TGF$\beta$

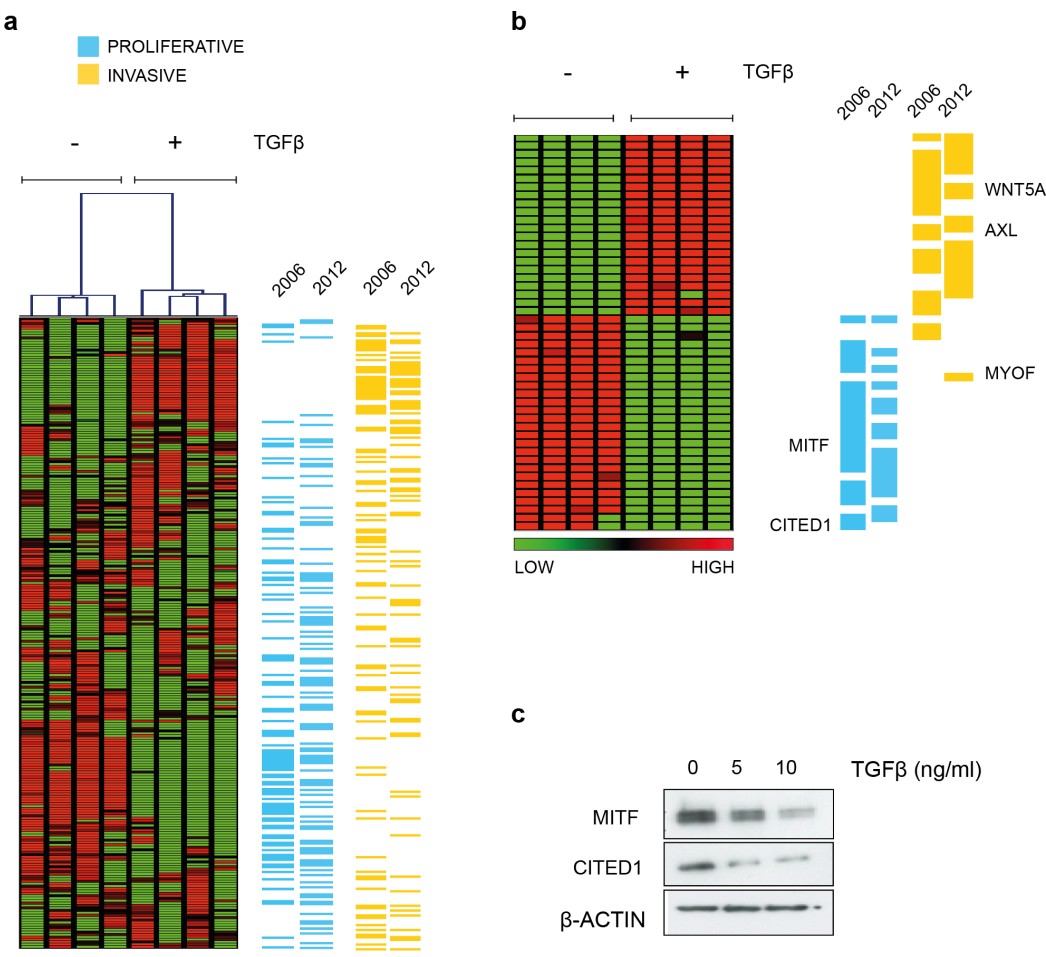

**Figure 1** **TGFβ induced gene expression in A2058 melanoma cells.** (A) Distribution of the proliferative and invasive signature score genes relative to the heatmap of gene expression changes induced by TGFβ treatment. Vertical lanes in the hierarchical cluster represent TGFβ treated or untreated replicates in the SAM 2-group comparison (B) Gene expression heatmap of the proliferative and invasive signature within those genes significantly altered by TGFβ treatment (1009 transcripts following SAM, median FDR $q$-value = 1%, the full list can be found in File S1). '2006' refers to the original signature list (motif1 and motif2, see *Hoek et al., 2006*) while '2012' refers to the updated signature derived from further datasets (*Widmer et al., 2012*). These lists can also be found in Data S1. (C) Western blot of MITF and CITED showing both proteins are suppressed by TGFβ treatment. *β*-Actin is used as a loading control.

treatment resulted in both upregulation of invasive signature genes and suppression of genes characterizing the proliferative phenotype using a SAM two-group comparison (Fig. 1A and Material & Methods section). The effect is most pronounced if only those signature genes that were deemed significantly altered by TGFβ treatment are examined. The original signature set defined by Hoek et al., was redefined as more public datasets became available and has a slightly different but overlapping gene profile based on the top ranked differentially expressed genes (Fig. 1B). Both MITF and CITED1 are in the proliferative cohort and their response to TGFβ treatment was confirmed at protein level in A2058 cells (Fig. 1C).

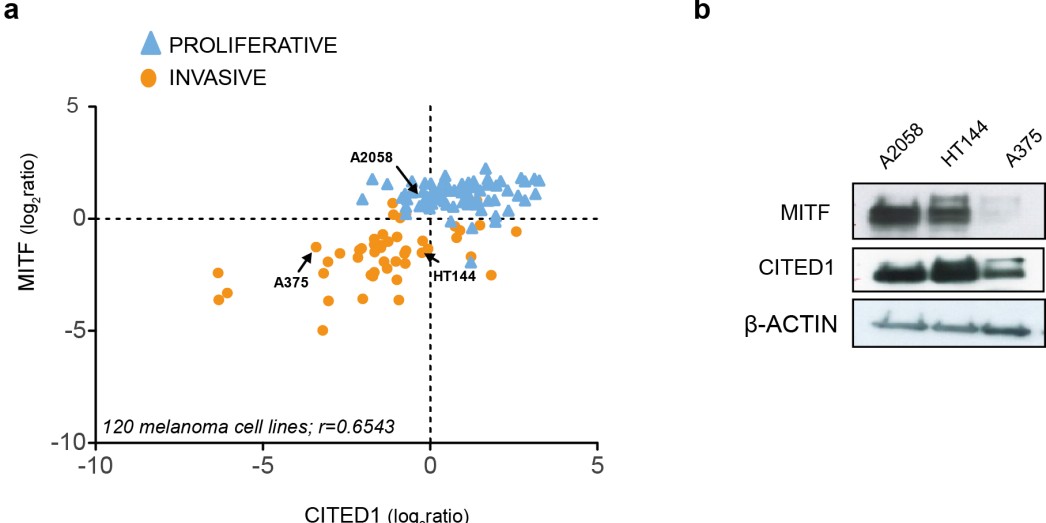

**Figure 2 CITED1 expression correlates with MITF expression.** (A) The relative MITF and CITED1 expression levels from the publicly available gene expression data of 120 melanoma cell lines (Pearson correlation $r = 0.6543$, $p < 0.001$). The full list of cells lines and expression data can be found in Data S2. Arrows indicate the cell lines used in this study. The cell lines are further subdivided into one of either 'invasive' of 'proliferative' phenotype based on expression signature score. (B) A Western blot is shown of the relative protein expression levels of both MITF and CITED1 in our cell lines in good agreement with the transcript levels.

## CITED1 expression positively correlates with the expression of MITF

Examination of publically available gene expression data on 120 melanoma cell lines demonstrated a consistent positive correlation between CITED1 and MITF expression ($r = 0.6543$). Each cell line was assigned as either 'proliferative' or 'invasive' based on a score derived from the averaged expression values of the approximately 50 genes in each defining signature set that had matching gene symbols in our data (Fig. 2A). We also confirmed the correlation in cell lines derived from our own lab (Fig. S1). This was important as inconsistency in interlaboratory phenotype signatures has previously been reported (*Widmer et al., 2012*). We could additionally confirm expression at the protein level (Fig. 2B).

## Gene expression analysis reveals CITED1 silencing can induce a phenotype-switch

To investigate the function of CITED1 in melanoma, we transiently downregulated its expression using CITED1 targeting siRNA. We choose the HT144 cell line as it had a relatively high level of detectable CITED1 mRNA and protein expression. A scatterplot of the 120 cell lines assigned as either 'proliferative' or 'invasive' based on the maximum matching gene signature score, demonstrates the shift in phenotype that occurs following CITED1 downregulation (Figs. 3A and 3B). A heatmap of the expression profiles clearly illustrates that the shift is due to a general induction of the 'proliferative' and suppression of the 'invasive' cohort (Fig. 3C). It was apparent that the #3 siCITED1 siRNA was not as effective at switching the cells as the #1 siCITED1, this was observed consistently

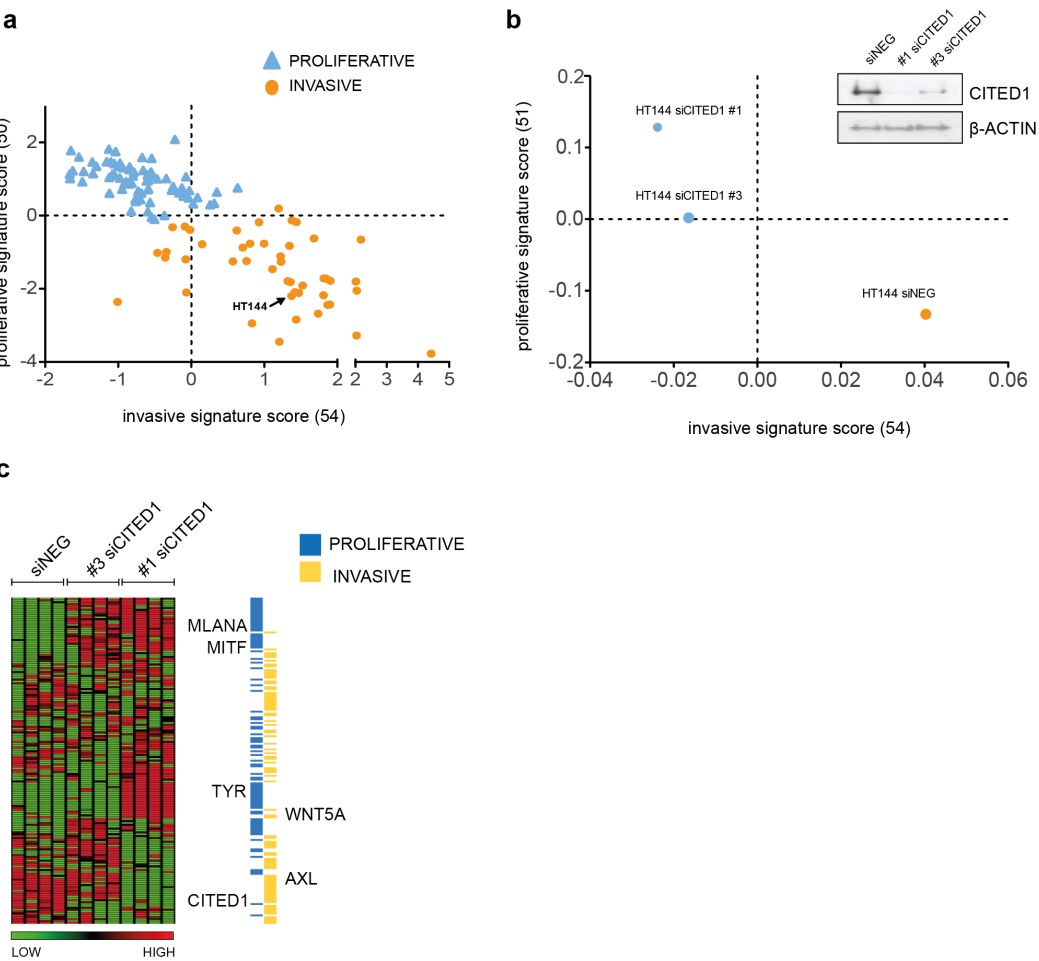

**Figure 3** **CITED1 silencing induces a phenotype switch.** (A) 120 melanoma cell lines are shown distributed on the basis of the phenotype score. The HT144 cells chosen to study the effects of CITED1 downregulation are indicated. (B) Following CITED1 downregulation a phenotype shift is observed indicated by their scatter position change according to the average expression score of genes that distinguish invasive from proliferative phenotype. For the 120 melanoma cell lines (Affymetrix platform) the expression score was derived from expression levels of 50 and 54 proliferative and invasive genes with matching genes symbols, respectively, while for the HT144 experiment (Illumina platform), 51 and 54 proliferative and invasive genes with matching genes symbols were retrieved. A Western blot of the degree of protein downregulation of CITED1 at the time of the expression analysis is also shown. $\beta$-Actin is used as a loading control (inset). (C) A heatmap comprising the 'invasive' and 'proliferative' signature genes illustrating how they are altered by CITED1 silencing; #1 and #3 denote two separate siRNAs targeting CITED1 and the vertical lanes represent the 4 replicates per treatment.

throughout our experiments and may be due to the fact that #3 siCITED1 was not as successful at silencing CITED1 (Fig. 3B, *inset*).

## CITED1 is a reciprocal regulator of MITF and impacts MITF target gene expression

A heatmap highlights the identity of only the significantly differentially induced transcripts between siNEG and both #1 & #3 siCITED1 (Fig. 4A). Of most relevance, we found MITF,

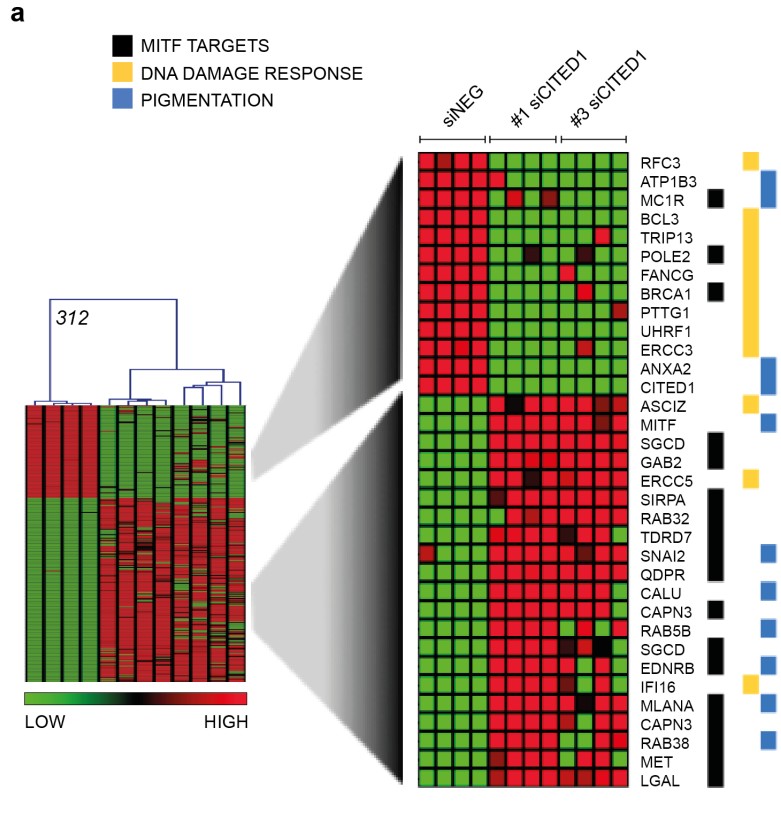

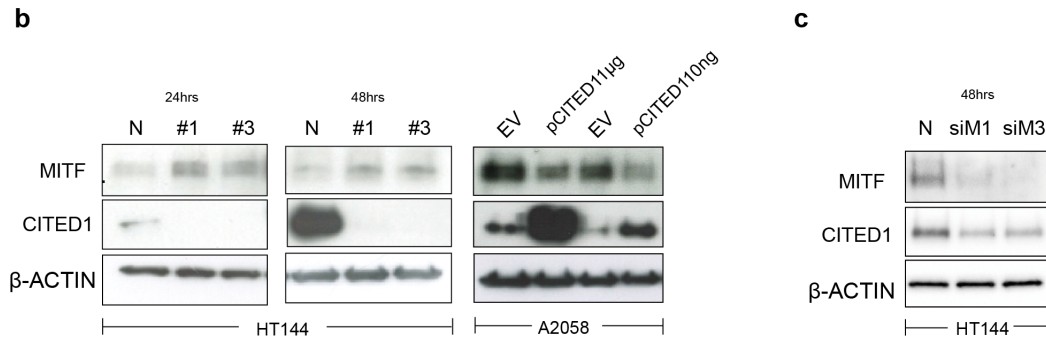

**Figure 4  CITED1 regulates MITF and its targets genes.** (A) A heatmap showing the 312 transcripts identified as significantly changed using a SAM 2-way comparison between siNEG and siCITED1 (#1 & #3 were combined), median FDR $q$-value = 3%. The four replicates from each condition, siNEG and, siCITED1#1 and siCITED1#3, respectively, group together in the hierarchical cluster shown. The full list of genes can be found in File S2. MITF, as well as a cohort of significantly enriched MITF targets, genes associated with pigmentation, and genes involved in the UV/DNA damage response are highlighted on the right. (B) Western blot confirmation of the effect of silencing CITED1, using siRNA (#1, #3) relative to a negative control siRNA (N), on MITF protein expression in HT144 cells at 24 and 48 h post-transfection, and the effect of overexpressing CITED1 (pCITED1) relative to an empty vector control (EV) in A2058 cells at 24 h post-transfection. $\beta$-Actin is used as a loading control in each case. (C) Western blot showing the effect of silencing MITF using two siRNAs (siM1, siM3) on both MITF and CITED1 levels in HT144 cells at 48 h post-transfection relative to a negative control siRNA (N). $\beta$-Actin is used as a loading control.

a known driver of the proliferative phenotype switch and many of its previously known downstream targets, these also included genes categorized by Gene Ontology annotation (GO) as related to pigmentation and UV/DNA damage response (Fig. 4A) (*Hoek et al., 2008b*; *McGill et al., 2006*; *Sánchez-Martín et al., 2002*; *Strub et al., 2011*). We could confirm that indeed MITF protein levels were affected by siCITED1 in HT144 cells and that conversely, overexpression of CITED1 in A2058 cells, resulted in downregulation of MITF (Fig. 4B). *Strub et al. (2011)* identified a large number of genomic targets of MITF by ChIP-seq analysis. A comparison of the genes differentially expressed by siCITED1 compared to siNEG, revealed that there was significant enrichment of these potential targets (Fig. S2A). Notably, genes both up and down regulated by siCITED1 are represented among genes defined as having MITF-occupied promoters (Fig. S2B). We also found that downregulation of MITF using siRNA in HT144 cells (Fig. 4C) and in WM293A, and SKMEL5 cells (Figs. S3A and S3B) resulted in decreased protein expression of CITED1 suggesting reciprocity between these factors.

## Induction of MITF by CITED1 silencing transiently restrains cell cycle progression and impacts cell viability

To investigate the effect of CITED1 silencing on melanoma cells behaviour we analysed the cell cycle distribution following siRNA treatment by flow cytometry. In siCITED1 treated HT144 cells we saw G1 accumulation as indicated by an increase in the diploid G1 fraction and a concomitant reduction in the total S-phase fraction peaking at 33 h but also observed at 48 and 72 h post-transfection in comparison to siRNA control HT144 cells. Again, the effect was apparent but not as pronounced using the #3 siCITED1 (Fig. S4A). Similar effects were seen in #1 and #3 siCITED1 treated A2058 and A375 cells (Figs. S4B and S4C).

Owing to the previously reported dependency of MITF-induced cell cycle arrest on CDKN1A/P21 we investigated the levels of several cyclin-dependant kinase inhibitors following CITED1 silencing (*Carreira et al., 2005*). We found that CDKN1A/P21 was transiently increased in siCITED1 treated HT144 cells relative to the siNEG treated HT144 cells (Fig. 5B). In contrast, in A2058 cells, which do not have detectable levels of CDKN1A/P21 (Fig. S5), the levels of CDKN1C/P57 were supressed in response to CITED1 overexpression (Fig. 5B). We hypothesised therefore that melanoma cells can utilise either CDKN1A/P21 or CDKN1C/P57 to mediate cell cycle arrest induced by MITF and this is reflected in the expression levels of the alternate CDK inhibitors in different melanoma cell lines (Fig. S5).

In agreement with the cell cycle data, an Alamar Blue assay revealed a significant reduction in cell viability as measured by metabolic activity over 5 days in HT144 cells treated with siCITED1 (Fig. 5C). The effect was apparent but not as pronounced in the #3 siCITED1 sample.

## The effect of CITED1 silencing on MITF is transient and mediated via promoter activation

We observed that the peak upregulation of MITF and CDKN1A/P21 protein following siCITED1 treatment varied from transfection to transfection, being seen between

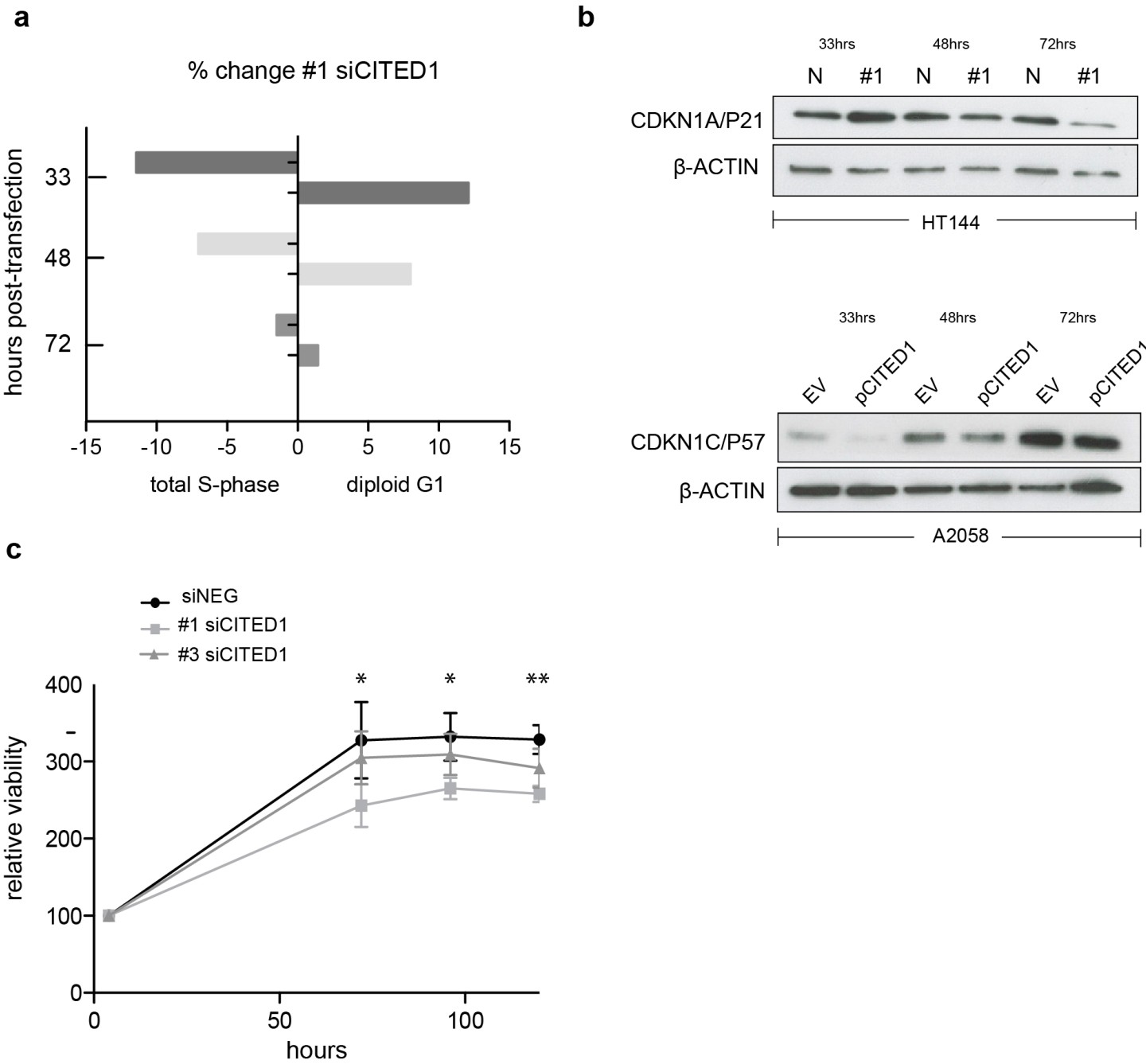

**Figure 5 CITED1 silencing restrains cell cycle progression and reduces cell viability.** (A) A bar chart showing the % change in cell cycle distribution in #1 siCITED1 treated HT144 cells relative to siNEG treated HT144 cells. The reduction in the total S-phase is shown at 33 h, 48 h and 72 h post-transfection in addition to the corresponding increase in the diploid G1 fraction. (B) Western blots showing upregulation of CDKN1A/P21 following CITED1 silencing in HT144 cells and suppression of CDKN1C/P57 following CITED1 overexpression in A2058 cells. (C) An Alamar Blue based metabolic assay shows a reduction in cell viability over 5 days in HT144 cells treated with siCITED1 relative to those treated with siNEG. Stars indicate significance for siNEG vs. #1 siCITED1 where *** $p <= 0.0005$, ** $p <= 0.005$ and * $p <= 0.05$. In the case of siNEG vs. #1 siCITED1, the difference is significant (*) at 96 and 120 h.

24–48 h post-transfection but appearing as unchanged or even downregulated after this time (Fig. 6A). In agreement, later timepoints of the cell cycle analysis (=/>72 h) exhibited little or no change in G1/S-phase distribution or even a reverse pattern (Fig. 5A HT144, and data not shown: A2058, A375). We therefore sought to examine the transcriptional dynamics more closely, map the changes in MITF following CITED1 silencing and see if they corresponded to cell behaviour and changes at the protein level. We used a quantitative droplet digital PCR based assay (Bio-Rad) to measure mRNA in HT144 cells transfected with siCITED1#1 and siNEG as well as A2058 cells transiently overexpressing CITED1 compared to an empty vector control. MITF, CITED1 and IPO8 specific primers and probes were used to measure exact copies/μl of each mRNA from aliquots of the same cDNA solution. Plots of siCITED1(copies/μl)/siNEG(copies/μl) and EV(copies/μl)/pCITED1(copies/μl) show the directional change in MITF and CITED1 relative to the housekeeper IPO8. CITED1 expression is rapidly supressed following siCITED1 treatment of HT144 cells, concomitant with an upregulation of MITF that diminishes over time and in fact is supressed by 100 h in accordance with observations at the protein level (Figs. 6A and 6B). In contrast, overexpression of CITED1 in A2058 cells results in transient suppression of MITF at both protein and transcript level (Figs. 6C and 6D).

The rapid MITF transcriptional response to CITED1 manipulation suggested to us that the effect could be directly mediated at the promoter level. To test this hypothesis, we over expressed an MITF-M promoter-reporter construct and CITED1 in A375 cells. We chose A375 cells, as while they had less endogenous CITED1 and MITF than HT144 or A2058 so as not to cause interference with the assay, we also knew that they could respond adequately as they had an identical G1 accumulation/S-phase decrease to both HT144 and A2058 cells following CITED1 silencing (Fig. S4C). TGFβ treatment was used as a positive control for repression of the MITF-M promoter. CITED1 transfection led to significant suppression of the MITF-M promoter luciferase activity relative to the empty vector control, as did TGFβ treatment alone or combination with CITED1 overexpression (Fig. 6E). There did not appear to be an additive or synergistic effect using both TGFβ treatment and CITED1 overexpression suggesting TGFβ may be dependent on CITED1 for MITF suppression.

## The CITED1-silenced gene signature predicts outcome in primary melanoma

The 'proliferative' and 'invasive' signature phenotypes have served to define the gene expression classification of melanoma cell lines. However, primary tumours and metastatic lesions have also been molecularly classified into several distinct groups by gene expression profiling (Harbst et al., 2012; Jonsson et al., 2010). The four-class structure found in tumours consists of the 'pigmentation,' 'proliferative,' 'high-immune' and 'normal-like' subgroups with a subset falling into an unclassifiable cohort (Jonsson et al., 2010). We used the same tumour classification to subtype the 120 cell lines that had publically available expression data and could show that the tumour 'pigmentation' subgroup that highly expresses MITF, corresponds to the cell line 'proliferative' phenotype described by Hoek et al. Accordingly, the tumour 'proliferative' and 'high-immune' subgroups

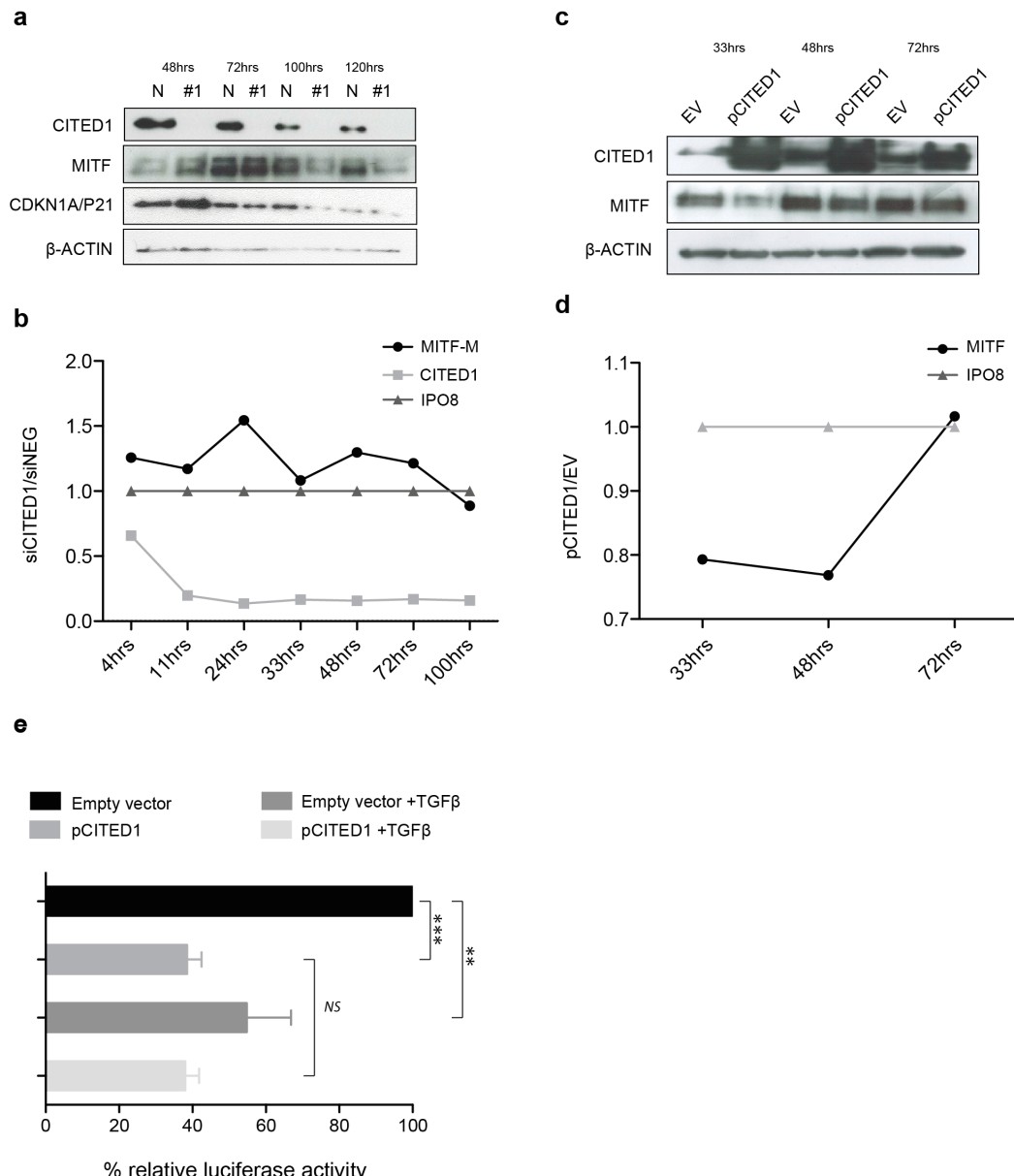

**Figure 6 CITED1 silencing transiently upregulates MITF via promoter activation.** (A) A Western blot of HT144 cell lysate samples taken at the indicated time points post-transfection and showing the corresponding levels of MITF protein in #1 siCITED1 and siNEG treated cells. (B) The changes in mRNA levels of MITF-M, CITED1 and a housekeeper gene IPO8, as measured by specific ddPCR assays over a time course of 4–100 h following transfection of HT144 cells with either siCITED1 or siNEG. (C) A Western blot of A2058 cell lysate samples taken at the indicated time points post-transfection and showing the corresponding levels of MITF protein between CITED1 overexpression (pCITED1) and empty vector (EV) control. (D) The changes in mRNA levels of MITF and a housekeeper gene IPO8, as measured by specific ddPCR assays over a time course of 33–72 h following transfection of A2058 with either pCITED1 or an empty vector control. (E) The relative luciferase activity of the MITF-M promoter reporter measured in lysates of A375 cells transfected with the pCITED1 expression plasmid or empty vector (EV) control and treated with or without TGF$\beta$ for 24 h (*** $p <= 0.0005$, ** $p <= 0.005$ and NS = not significant).

comprise the cell line 'invasive' phenotype (Fig. 7A). It is worth noting that the names of the tumour subgroups were derived from a description of the differentially expressed genes that comprised each molecular classification while the 'invasive-proliferative' switching phenotypes were named to reflect the *behaviour* exhibited by lines classified by one or other signature. This distinction helps to explain the confusing occurrence that both classifications have a group referred to as 'proliferative' although they are not equivalent.

The overlap between the primary tumour classifying and cell line classifying systems allows us to infer that CITED1 expression is most likely restricted to a subset of MITF high 'pigmentation' subtype tumours. As the tumour subtype classification was shown to be prognostically significant in primary melanomas we were interested to know if CITED1 expression itself was independently predictive of outcome. Previously we reported on the analysis of 223 primary lesions using the Illumina WG-DASL protocol (*Harbst et al., 2012*). As the CITED1 probe in this assay did not produce reliable data we instead derived a CITED1-silenced gene signature score based on the differentially expressed genes from the HT144 siCITED1 experiment (Fig. 3). We therefore effectively created a multi-gene surrogate expression signature rather than using CITED1 gene expression itself. We subsequently interrogated the gene expression data on the primary melanoma lesions using a nearest centroid approach derived from the CITED1-silenced gene signature. This revealed that primary melanomas with a gene expression signature most similar to the CITED1-silenced signature (CITED1low-class) had a significantly better outcome than those with a signature most disparate from the CITED1-silenced signature (CITED1high-class) (Fig. 7B). Importantly, the CITED1 signature classing had independent prognostic information (HR 1.85, CI [0.30–0.98], $p = 0.044$) from the AJCC staging system (HR 5.05, CI [2.42–10.55], $p = 1.64 \times 10 - 5$). Accepting the caveat that we depend here on a proxy gene-signature, these data indirectly imply that CITED1 expression is a potential prognostic indicator in primary melanomas and the transcriptional program influenced by CITED1 expression determines tumour behaviour *in vivo*.

## DISCUSSION

One seemingly paradoxical observation from our study and previous investigations is that although CITED1 behaves as a negative regulator of MITF, both their expression levels appear positively correlated across cells lines and tumours. We maintain that this observation simply reflects the fact that where there are high levels of MITF, high levels of its negative regulator are also required. The evidence of the tight control exerted over MITF levels in melanocytes and melanoma simply speaks to the necessity of the cell to maintain a level compatible with survival and proliferation, in a type of biological 'sweet-spot' facilitating tumour progression. The cellular effects of both extremes i.e.,: very low or high levels of MITF, have been elegantly described by a rheostat model in order to reconcile the conflicting observations of the effects of manipulating MITF *in vitro*, and the fact that counter-intuitively, a lineage-specifying differentiation factor can behave as a potent oncogene (*Hoek & Goding, 2010*; *Carreira et al., 2006*; *Cheli et al., 2011a*; *Cheli et al., 2011b*). The rheostat model (Fig. S6) attempts to explain why MITF silencing can block

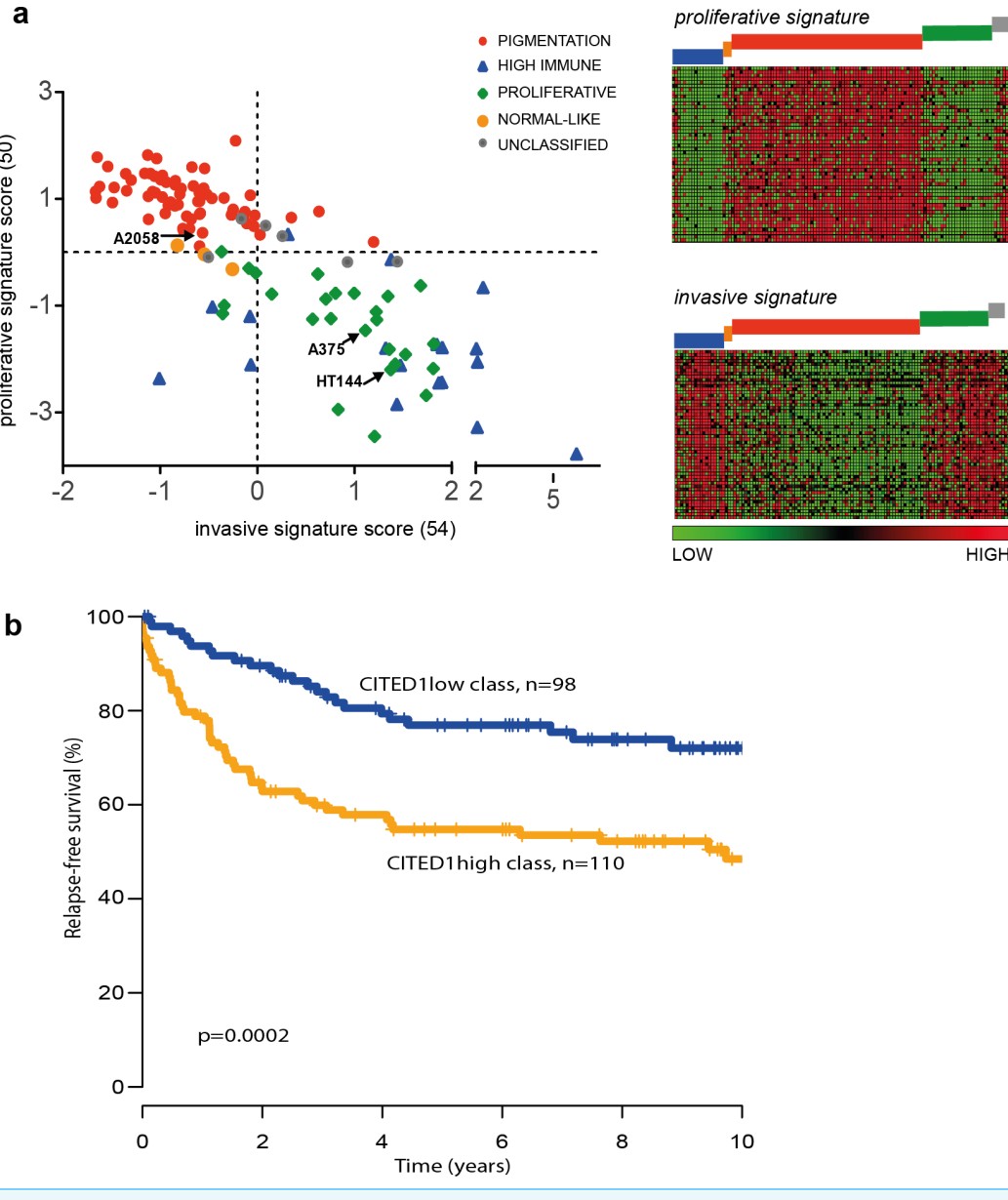

**Figure 7 The CITED1-silenced gene signature predicts patient outcome.** (A) In the leftmost panel A scatter plot of the 120 melanoma cell lines are shown distributed on the basis of their 'invasive' or 'proliferative' phenotype signature score and coloured according to the tumour molecular subtypes as defined by *Jonsson et al. (2010)* to illustrate the overlap between the two classification systems. In the rightmost panel the data is presented as a heatmap where each gene of the proliferative or invasive signature genes is represented by a horizontal line and the 120 individual cells lines are grouped by molecular tumour subtype (coloured blocks) and shown vertically. (B) Recurrence free survival (RFS) of primary melanoma patients grouped by gene expression similarity to the CITED1 (siCITED1) silenced gene signature.

cells in G1 and induce senescence, while it is also possible to induce a G1 arrest by MITF overexpression via CDKN2A/P16 or CDKN1A/P21 and, as we now propose, potentially also via CDKN1C/P57 (*Carreira et al., 2006*; *Loercher et al., 2005*; *Carreira et al., 2005*). At the extreme high end of MITF expression lies differentiated melanocytic cells, while the lowest levels can lead to senescence and irreversible cell death. Between these two extremes however it is thought that melanoma cells can oscillate from a low-MITF 'invasive' to a high-MITF 'proliferative' state via phenotype-switching.

We hypothesise that the role of CITED1 in melanoma is to maintain levels of MITF compatible with tumour progression and effectively tip the balance in favour of cell cycle progression rather than MITF-induced G1-arrest. This is supported by our findings that downregulation of CITED1 using siRNA results in a phenotype switch to a more pigmented state driven by increased MITF expression and concomitant upregulation of CDKN1A/P21. Conversely, we could observe that downregulation of MITF resulted in suppression of CITED1 in several cell lines suggesting the existence of a classical feedback loop where low MITF levels result in inhibition of its negative regulator. MITF induced cell cycle arrest was previously shown to be dependent on CDKN1A/P21 and it was demonstrated that MITF does not cause a cell cycle arrest in CDKN1A-deficient mouse embryo fibroblasts (MEF) cells (*Carreira et al., 2005*). However, our data indicate that in melanoma cells deficient in CDKN1A/P21, the alternative CDK inhibitor CDKN1C/P57 is expressed and responsive to MITF.

Interestingly, while we observed upregulation of most MITF targets following CITED1 silencing, we found that BRCA1 and other DNA damage response (DDR) genes were supressed, suggesting that CITED1 downregulation does not necessarily facilitate transcription of all MITF targets. It is thus tempting to speculate that rather than simply acting to induce MITF and thereby indirectly enhance transcription of its target genes, that CITED1 may also act as co-factor for MITF at various genomic locations differentially modulating the MITF target gene response at individual promoters. One way that this might be achieved is via MITF-CITED1 competition for CBP/P300 binding as CBP/P300 is a known transcriptional coregulator for MITF, although it is not required for transcription of all MITF targets (*Vachtenheim, Šestáková á & Tuháčková, 2007*; *Yan et al., 2013*).

As suggested by Sáez-Ayala et al., anti-cancer therapy should be ideally independent of dominant or 'driver' genetic alterations so that subclonal populations do not gain a subsequent advantage and the same holds true in the case of targeting a specific phenotype. Successful therapy will necessarily need to switch or push the subdominant phenotype into the susceptible state or eradicate the phenotype resistant to treatment. This approach was initially championed by *Cheli et al. (2011a)* and *Cheli et al. (2011b)*, who proposed the eradication of low-MITF cells as a therapeutic strategy. Indeed the idea of lineage-specific therapy has been subsequently proved in principle using methotrexate (MTX) to first activate MITF expression, in turn activating the tyrosinase enzyme, and thereby sensitising tumour cells to a tyrosinase-processed anti-folate prodrug (TMECG) (*Sáez-Ayala et al., 2013*). However, even without drug targeting, induction of MITF, to levels seen in melanocytes or above what is tolerated by even the highly pigmented

tumour cell types, would seem to be incompatible with melanoma progression as it can inhibit cell cycle progression (*Goding, 2013*). Our assertion is that CITED1 acts to repress MITF in order to maintain its level in a range compatible with tumourigenesis. This assertion as a consequence naturally suggests CITED1 as therapeutic target for genetic manipulation. Successful implementation of such a strategy would result in cell specific enhancement of MITF expression and increased susceptibility to the type of chemotherapeutic eradication demonstrated by Sáez-Ayala et al. or potentially induction of CDKN1A/p21 or CDKN1C/p57-dependent cell growth arrest even without further intervention (Fig. S6) (*Sáez-Ayala et al., 2013*).

## ACKNOWLEDGEMENTS

We would like to thank Dr. Claudia Wellbrock (University of Manchester, UK) for the kind gift of the PG2-MITF promoter-reporter construct and Professor Toshi Shioda (MGH, Harvard) for the human CITED1 pRc/CMV expression vector. We also would like to acknowledge Professor Bo Baldetorp (Division of Oncology, Lund University) for his advice on cell cycle analysis and the support of Professor Tommy Andersson (Cell and Experimental Pathology, ILMM, Lund University).

### Funding

This work was supported by Mrs. Berta Kamprad Foundation (JH, SGS, GJ, AB), Swedish Research Council (GJ, AB), Gunnar Nilsson Cancer Foundation (GJ, AB), Swedish Cancer Society (GJ, AB & Tommy Andersson [CAN 2011/726]), and The Gustav Vth Jubileefoundation (GJ) and BioCARE (GJ, SGS). The funders had no role in study design, data collection and analysis, decision to publish, or preparation of the manuscript.

### Grant Disclosures

The following grant information was disclosed by the authors:
Berta Kamprad Foundation.
Swedish Research Council.
Gunnar Nilsson Cancer Foundation.
Swedish Cancer Society: CAN 2011/726.
The Gustav Vth Jubileefoundation.
BioCARE.

### Competing Interests

They authors declare there are no competing interests.

### Author Contributions

- Jillian Howlin conceived and designed the experiments, performed the experiments, analyzed the data, wrote the paper, prepared figures and/or tables, reviewed drafts of the paper.
- Helena Cirenajwis performed the experiments, reviewed drafts of the paper.

- Barbara Lettiero performed the experiments.
- Johan Staaf analyzed the data, contributed reagents/materials/analysis tools, reviewed drafts of the paper.
- Martin Lauss analyzed the data, reviewed drafts of the paper.
- Lao Saal contributed reagents/materials/analysis tools.
- Åke Borg contributed reagents/materials/analysis tools, reviewed drafts of the paper.
- Sofia Gruvberger-Saal and Göran Jönsson conceived and designed the experiments, analyzed the data, contributed reagents/materials/analysis tools, reviewed drafts of the paper.

## Microarray Data Deposition

The following information was supplied regarding the deposition of microarray data:

The data is MIAME compliant.

Public data was used from PMID: 17516929, 16827748 and 20406975.

GEO accession numbers:

http://www.ncbi.nlm.nih.gov/geo/query/acc.cgi?acc=GSE66113

http://www.ncbi.nlm.nih.gov/geo/query/acc.cgi?acc=GSE66114.

## Supplemental Information

Supplemental information for this article can be found online at http://dx.doi.org/10.7717/peerj.788#supplemental-information.

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
