# Peer review of "Loss of CITED1, an MITF regulator, drives a phenotype switch in vitro and can predict clinical outcome in primary melanoma tumours"

_PeerJ, doi:10.7717/peerj.788_

## Round 0.1 · original submission · Major Revisions

Please address the reviewers' comments.

Reviewer 1 ·

Basic reporting

Issues pertaining to the Abstract and Introduction:

1. Abstract: “CITED1 expression is repressed by TGFB in addition to other ‘proliferative’ signature genes while invasive genes are upregulated?” This sentence is unclear. As I read it this sentence indicates that TGFB and other ‘proliferative’ genes are repressing CITED1. But I think the authors mean that TGFB represses CITED1, and that TGFB also represses other proliferative genes. Please re-word.
2. Intro: Line 19; The authors state that B16F10 are non-pigmented, this is not true, they do have pigmentation although it may be weaker than B16F1 (as is stated in the Shioda reference they cite). Please correct.
3. Intro and rest of text general: Some editing needed of spelling and grammar (lack of space between last word and reference line 21 and 25). Again line 219.
4. Line 69 typo: Target MITF genes include MCIP – do the authors mean MC1R?

Experimental design

Issues relating to the Methods:

5. Methods: Not all of the company suppliers are listed for reagents (eg media)
6. Gene expression analysis line 110: The authors state that 4 replicates were used for each treatment. Were experimental replicates analysed (eg cells at a different passage on a different day) or were these technical replicates within the one experiment (eg different wells from the same plate)? Please add more detail.
7. Typo line 137 (manufactures).
8. Line 148: Has the CITED1 construct been used before? If so please provide a reference, if not perhaps the authors could provide sequencing data? Or at least details of how the sequence was checked?

Validity of the findings

Issues relating to the Results and Discussion:

9. Results: Line 214: The authors state that MITF levels are increased and that invasiveness is enhanced in response to TGF. However in the articles they cite MITF levels are actually decreased in response to TGF. Please correct.
10. Although In Figure 1B it is clear that for genes where expression is deemed “significantly altered” by TGFB the proliferative genes go down with TGF and the invasive genes go up, in Figure 1A there do look to be some proliferative genes that go up with TGF treatment and some invasive genes that go down. Although it would be difficult to display all the gene names in the heat map it would be good to see the full list of ‘proliferative’ and ‘invasive’ genes to see which particular genes are significantly altered (up or down), and also to see genes that were not considered ‘significantly altered’ by TGFB. Perhaps as a supplementary excel file? Also in the methods and figure legend it says that 1009 probes were significantly altered by TGFB, I assume this is including genes that are not in the ‘proliferative’ or ‘invasive’ signature groups, perhaps the full list of these genes could also be provided in supplementary data? It would also be good to have the full list of genes for the other microarray figures with CITED1 siRNA.
11. I could not find any Figure legends for the supplementary Figures? Could the authors indicate the points that represent the same cell lines in Figure 2A that are used in supplementary Figure S1 so the levels can be compared?
12. Line 294: I think there should be a reference to Figure 5b here?
13. General note for Figure legends with western blots: Can the reviewers please indicate how many experiments these western blots are representative of.
14. Discussion: Line 452: The authors state that the expression of MITF levels higher than that seen in melanocytes would be incompatible with melanoma progression as it inhibits cell cycle progression. Actually MITF levels can be similar to or even higher than those seen in primary melanocytes (eg Cook et al, Exp Cell Res, 2005; GEO Microarray dataset GDS1965) and this does not seem to inhibit proliferation. In addition amplification of MITF may act as an oncogene. Can the authors comment on this.

Additional comments

Overall this is a well-written paper with well-controlled experiments. However, there are a number of minor issues that need addressing.

Reviewer 2 ·

Basic reporting

This manuscript by Howlin et al describes the effects of CITED1 in melanoma and its relationship to MITF. The manuscript is mostly well written and the figures clearly constructed, although details are missing in some cases (see below). The data presented is indicative of a relationship between CITED 1 and MITF, although many questions still remain. For example, is CITED1 affecting MITF protein stability? It is known that MITF affects its own regulation and the effects of CITED 1 do not need to be direct. Similary, the data showing possible prognostic value of CITED1 are not very strong.

Experimental design

Comments
1. Figure 1, although visually clear and pleasing, does not provide enough information for understanding what is being shown. For example, what do the different lanes in the heatmap indicate (presumabley replications?)? Why are the genes involved not indicated? In panel b, only some of the genes are indicated. Why not all? The legend says that „´2006´ refers to the signature list (motif1 and motif 2)“. Which lists are these?
2. Figure 2 examines 120 cell lines. Which lines are these exactly?
3. In Figure 3, the authors claim that they observe a shift in phenotype upon siCITED1. How was the phenotype determined? Was this only through effects on gene expression and if so, were statistics used to get a handle on significance?
4. In Figure 4a, it is not clear what the lanes indicate.
5. In Figure 4b, the effects of siCITED1 on MITF are not very impressive. The control band is faint and so are the experimental bands. Did this replicate in subsequent experiments? How many replicates were performed? Were the bands quantitated to determine statistical significance?
6. In Figure 5c, the y-axis is labeled „relative viability“ whereas the text says „metabolic activity“. Which is correct?
7. In Figure 5b, the p21 band goes up after siCITED1 after 33 hrs but then goes down. This is also seen in Figure 6a but in this figure the p21 levels are high after 48 hrs. Why is this difference with time? Were these effects quantitated and replicated?
8. In Figure 6a, the effects on MITF mRNA seem limited. The value hovers around 1.25 in all the samples except the last one. The authors therefore overstate the effects of siCITED on MITF mRNA. The authors should determine effects of siCITED on MITF protein stability/translation.
9. The authors use the siCITED gene signature as a proxy for CITED expression in the cell lines. This is a leap of faith.

Validity of the findings

See above.

Additional comments

See above

---

## Round 0.2 · Minor Revisions

There are few minor revisions you may find useful to make in your manuscript.

Reviewer 1 ·

Basic reporting

No comments

Experimental design

No comments

Validity of the findings

Minor Issues:
1. The authors have added the Supplemental data S7 and S8 to supply more information about the gene expression changes. However, the genes are only listed and there is no data about whether these genes were upregulated or down regulated. Could this data be added to the files? Otherwise will this data be added to the GEO publically available database?
2. Western blotting: The authors state that they have only performed western blots in some cases once – but in other cases they deemed more important to repeat have been repeated several times to ensure reproducibility. This is ok – however I think that the exact number of times each western was repeated needs to be made clear in the figure legends so that anyone reading the manuscript is aware of the reproducibility of the findings. If there were differences depending on which time point was used (for example after siRNA knockdown), then this needs to be made clear (and if different time points cannot be combined then perhaps more repeats are necessary so that there are replicates for each time point). Densitometry should be performed in cases where the differences in protein expression are not obvious by eye (for example Figure 4b and S3).

Additional comments

Most of the my comments have been addressed sufficiently – however a couple of minor points remain (see comments on validity of the findings).

Reviewer 2 ·

Basic reporting

The manuscript is well written and clear. The authors have address all my concerns with the first submission.

Experimental design

The experimental design is solid.

Validity of the findings

The findings are interesting and important. The data is robust and statistically correct.

Additional comments

The authors have addressed all my concerns. Only two minor things that need to be addressed. 1. In line 268 the "of the only the significantly..." needs improvement. 2. in Figure 1 legend the "..untreated replicates in the SAM 2-group comparison" needs further explanation. Which SAM 2 comparison? This is the first mention of this group in the legends and there is no mention of it in the text describing Figure 1.

---

## Round 0.3 · accepted · Accept

There are no more comments.